# Architecture and mechanism of the late endosomal Rab7-like Ypt7 guanine nucleotide exchange factor complex Mon1–Ccz1

Stephan Kiontke[1], Lars Langemeyer[2], Anne Kuhlee[3], Saskia Schuback[1], Stefan Raunser[3], Christian Ungermann[2] & Daniel Kümmel[1]

The Mon1–Ccz1 complex (MC1) is the guanine nucleotide exchange factor (GEF) for the Rab GTPase Ypt7/Rab7 and is required for endosomal maturation and fusion at the vacuole/lysosome. Here we present the overall architecture of MC1 from *Chaetomium thermophilum*, and in combining biochemical studies and mutational analysis in yeast, we identify the domains required for catalytic activity, complex assembly and localization of MC1. The crystal structure of a catalytic MC1 core complex bound to Ypt7 provides mechanistic insight into its function. We pinpoint the determinants that allow for a discrimination of the Rab7-like Ypt7 over the Rab5-like Vps21, which are both located on the same membrane. MC1 shares structural similarities with the TRAPP complex, but employs a novel mechanism to promote nucleotide exchange that utilizes a conserved lysine residue of Ypt7, which is inserted upon MC1 binding into the nucleotide-binding pocket of Ypt7 and contributes to specificity.

[1] Division of Structural Biology, Department of Biology/Chemistry, University of Osnabrück, Barbarastraße 13, Osnabrück 49076, Germany. [2] Biochemistry Section, Department of Biology/Chemistry, University of Osnabrück, Barbarastraße 13, Osnabrück 49076, Germany. [3] Department of Structural Biochemistry, Max-Planck-Institute of Molecular Physiology, Otto-Hahn-Straße 11, Dortmund 44227, Germany. Correspondence and requests for materials should be addressed to D.K. (email: daniel.kuemmel@uos.de).

Eukaryotic cells are compartmentalized into organelles, which fulfil specialized functions. Exchange of substances between these membrane compartments is mediated by vesicular transport[1]. In this context, the endocytic compartment plays a central role in the sorting of cargo between the plasma membrane, the Golgi apparatus and the lysosome (vacuole in yeast) for recycling or degradation. Uptake of extracellular material and plasma membrane components into the cell is initially mediated by the budding of endocytic vesicles that fuse to form early endosomes (EEs) and subsequently mature to late endosomes (LEs). The LE acts as a general sorting station within the cell and represents a central hub in the endocytic pathway. Endosomal cargo can either be recycled to the Golgi or the plasma membrane, or is delivered to the lysosome/vacuole along with Golgi vesicles and autophagic structures for degradation.

The identity of different organelles is conveyed by Rab GTPases[2]. Rabs belong to the superfamily of Ras-like small GTPases and cycle between a GDP-bound 'off' state, in which they show cytosolic localization, and a GTP-bound 'on' state where Rabs associate with membranes and bind effectors. The nucleotide-binding pockets of Ras-like GTPases contain a guanine base recognition motif and P-loop motif, which coordinates the nucleotide β-phosphate and a $Mg^{2+}$ ion as essential cofactor. Furthermore, two variable regions - switch 1 and switch II - adopt distinct conformations depending on the nucleotide-loading state of the GTPase.

The intrinsic GTP hydrolysis rate of Rab GTPases is low; thus, their inactivation requires GTPase-activation proteins (GAPs). RabGAPs contain TBC (Tre-2/Bub2/Cdc16) domains, which insert conserved arginine and glutamine residues into the Rab nucleotide-binding pocket to catalyse GTP hydrolysis to GDP[3]. Activation of GTPases, which importantly coincides with membrane recruitment, in turn requires guanine nucleotide exchange factors (GEFs). In contrast to RabGAPs, RabGEFs are structurally and mechanistically diverse[4,5]. The underlying principle of nucleotide exchange common to all GEFs is that switch I and switch II are remodelled such that the affinity for nucleotide is lowered. GDP will leave the binding pocket and when the binding site is re-occupied by GTP, which is present at 10-fold higher concentrations than GDP in the cytosol, the transient GEF–GTPase complex falls apart. DENN domain proteins constitute the largest family of RabGEFs and remodel but do not block the nucleotide-binding pocket[6]. Vps9 GEFs, in contrast, additionally use insertion of a conserved aspartate to sterically expel nucleotide from the GTPase[7,8]. The TRAPP complex is a unique oligomeric GEF with three subunits participating in catalysis. As a variation to Vps9, TRAPP sticks an acidic glutamate into the nucleotide-binding pocket[9]. A fourth example for a RabGEF is Sec2, which forms an asymmetric coiled-coil dimer that has not been observed in any other GEF[10,11].

Finally, the family of heterodimeric RabGEFs consists of the Hps1–Hps4 (BLOC3) and Mon1–Ccz1 (MC1) complexes[12–14]. BLOC3 promotes nucleotide exchange for Rab32 and Rab38 and plays an essential role in the biogenesis of lysosome-related organelles[13]. Mutations in BLOC3 can cause the Hermansky-Pudlak syndrome, a genetic disease associated with albinism, defective cellular storage and bleeding disorders that leads to pulmonary fibrosis[15]. MC1 has been identified as the GEF for Rab7/Ypt7 and localizes to LEs[12,16]. The structure and mechanism of the heterodimeric RabGEFs is not known, but the presence of a longin domain in each subunit, which are required for complex formation, is a characteristic feature[12,13,17]. Longin domains are found in several Rab-interacting proteins, including DENN GEFs and the TRAPP complex, but the interacting structural motifs of the longin domain vary[18]. However, two longin subunits in TRAPP form a heterodimer that also contributes to GEF activity of TRAPP. A similar structural module might be used by the heterodimeric GEFs[9,17].

The MC1 complex has been characterized in several species[19–24]. It plays an essential role in a Rab cascade that defines the endolysosomal system and orchestrates the Rab switch between Rab5 and Rab7 (ref. 25). Its function is conserved in yeast, where it promotes the switch of Vps21 to Ypt7. EEs are initially positive for the GTPase Rab5 (Vps21 in yeast). The sequential inactivation of Rab5 and activation of Rab7 (Ypt7 in yeast), which then serves as marker for LEs, confers membrane identity and directionality during the endosomal maturation process[26]. The current model suggests that MC1 is initially recruited by Rab5/Vps21, the phospholipid phosphoinositol-3-phosphate (PI3P) and probably further factors to endosomes[27]. Via its GEF activity, MC1 will trigger localization of active Rab7/Ypt7 to the membrane. At the same time, MC1 is thought to oust the Rab5 GEF and promote recruitment of Rab5 GAP[28]. Thus, MC1 action will rapidly remove Rab5 from the endosomal membrane and replace it with Rab7, switching organelle identity from EE to LE.

The molecular mechanism, by which heterodimeric RabGEFs fulfil their functions, remained unclear. We now provide the structural and biochemical characterization of the MC1 complex architecture and the catalytic mechanism. Furthermore, we investigate the localization requirements for proper function of MC1 in vivo. Our study shows that MC1 forms a heterotetrameric complex with specific localization domains and a catalytic core that acts through a unique mechanism, thus representing a novel class of RabGEFs.

## Results

**Architecture of the Mon1–Ccz1 complex.** The heterodimeric Mon1–Ccz1 (MC1) complex from *Saccharomyces cerevisiae* (*Sc*MC1) has been successfully used in biochemical studies, but the recombinant proteins show low expression levels, aggregation and instability. Proteins from the thermophilic fungus *Chaetomium thermophilum* have been shown to be better suited for structural studies than homologues from other species[29]. We therefore used Mon1 and Ccz1 from *C. thermophilum*, which show an analogous domain architecture compared with the *S. cerevisiae* MC1 complex (Supplementary Fig. 1a): importantly, longin domains at the N-terminus of Ccz1 and in Mon1, which mediate complex assembly, are conserved. In addition, Mon1 and Ccz1 contain putative α-helical C-terminal domains, and the very N-terminal ~150 amino acids of Mon1 are predicted to be disordered.

The full-length *Ct*MC1 (*Ct*MC1full) complex could be produced using co-expression and was purified to homogeneity. The complex elutes as an apparent ~300 kDa particle with a 1:1 stoichiometry of *Ct*Mon1 and *Ct*Ccz1 on gel filtration, suggesting that the proteins form a dimer of heterodimers (Supplementary Fig. 1b). To more comprehensively investigate the overall architecture of the full complex, we used electron microscopy (EM). The *Ct*MC1 complex was analysed by negative stain EM and single particle analysis (Supplementary Fig. 1c–g). The complex has a globular ellipsoid structure with dimensions of 12 nm × 7 nm, consistent with a size range of 250–350 kDa. *Ct*MC1 lies on the grid at random orientations along the long axis of the particle, yielding different side views and intermediates between side and top views (Supplementary Fig. 1c). A total of 22,165 particles were classified into 100 classes (Supplementary Fig. 1d). The averages appear to be twofold symmetric, which indicates a dimerization of heterodimers. However, a three-

dimensional (3D) reconstruction at ~17 Å resolution (Supplementary Fig. 1e–g) of $Ct$MC1 does not show a clear twofold symmetry as one might expect, suggesting a non- or pseudo-symmetrical interaction. Overall, EM analysis establishes that $Ct$MC1 forms a tetramer with two copies of each protein, in line with the observations from gel filtration.

When co-expressed with a nucleotide-free mutant of Ypt7 ($Ct$Ypt7–N125I), a stable ~350 kDa complex consisting of two copies each of $Ct$Mon1, $Ct$Ccz1 and $Ct$Ypt7 was isolated on gel filtration (Supplementary Fig. 1h). We further dissected $Ct$MC1 architecture by generating truncation constructs. Removal of the $Ct$Mon1 N-terminus (residues 1–140) had no effect on complex stability; however, the additional deletion of $Ct$Ccz1 C-terminus (residues 250–796, $Ct$MC1Δ) eluted as an ~110 kDa particle containing MC1 and Ypt7. Thus, dimerization of the MC1 heterodimer was lost. We also co-expressed the predicted longin domain fragments ($Ct$MC1core: $Ct$Mon1 195–355 and $Ct$Ccz1 1–249). The longin domains had previously been shown to mediate interaction between Mon1 and Ccz1 (ref. 12). For the Mon1 construct, an additional N-terminal helix α0 was needed for stability of the protein. The core complex still bound Ypt7 but showed a trimer/hexamer equilibrium (Supplementary Fig. 1h). Because MC1Δ—Ypt7, which additionally contains the C-terminus of Mon1, elutes as a trimer, we conclude that hexamer formation of MC1core–Ypt7 is an artifact from the truncation of Mon1 that does not reflect a functional interaction mode and does not occur in the context of full complex. Taken together, the longin domains of Mon1 and Ccz1 form a hetero-dimer that is sufficient to bind Ypt7, and the C-terminus of Ccz1 represents a homodimerization domain for MC1.

**Requirements for MC1 complex functionality.** Binding studies showed that the $Ct$MC1core complex binds to Ypt7, suggesting that it might be sufficient to convey the catalytic activity of the whole complex. We used a fluorescence-based GEF activity assay[4] to compare the activity of $Ct$MC1full and $Ct$MC1core (Fig. 1a and Supplementary Fig. 2a,b). Recombinantly purified $Ct$MC1full showed a concentration-dependent GEF activity towards its Rab GTPase $Ct$Ypt7 upon addition of GTP ($k_{cat}/K_M$ 2.1 × 10$^4$ M$^{-1}$ s$^{-1}$). The enzyme-mediated exchange rates differed significantly from the intrinsic rate of the $Ct$Ypt7–MANT–GDP complex without $Ct$MC1. In contrast, absence of GTP induced no significant release of MANT–GDP, thus demonstrating that all observed nucleotide exchange reactions are GTP-driven (Supplementary Fig. 2a). The $Ct$MC1core complex exhibits a similar catalytic efficiency ($k_{cat}/K_M$ 2.3 × 10$^4$ M$^{-1}$ s$^{-1}$) compared with the full-length GEF complex, which demonstrates that residual parts are not crucial for GEF function.

Because the longin dimer subcomplex of MC1 was sufficient to catalyse nucleotide exchange of Ypt7, we wanted to address the function of the remaining domains of the complex. In yeast cells, MC1 has to localize to endosomal structures[12]. On the basis of our analysis of the architecture of $Ct$MC1, we designed truncations of yeast Mon1 and Ccz1 and tested their ability to rescue the vacuole fragmentation phenotype in a knockout background (Fig. 1b,c). The deletion of the N-terminus of Mon1 did not affect the localization of the protein and consequently rescued vacuolar morphology. In contrast, deletion of the Mon1 C-terminus alone and in combination with the N-terminus resulted in cytosolic localization and vacuoles remained fragmented. Similarly, deletion of the Ccz1 C-terminus rendered the complex dysfunctional, and the protein was mislocalized. This suggests that the C-terminal domains of Mon1 and Ccz1 jointly are required to localize

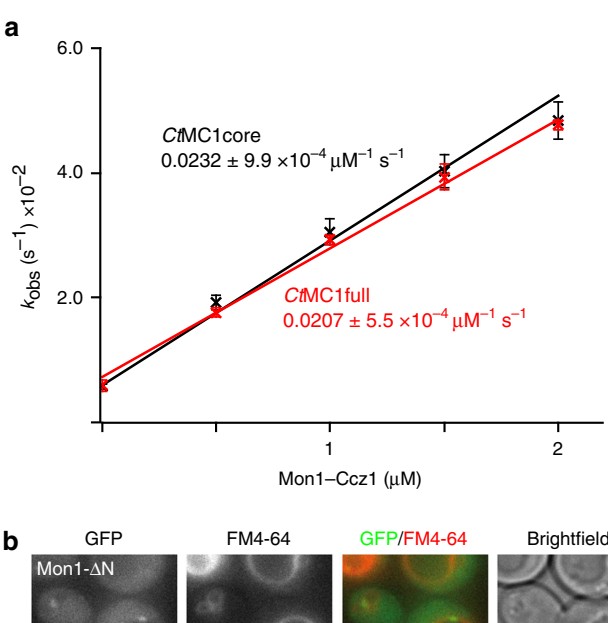

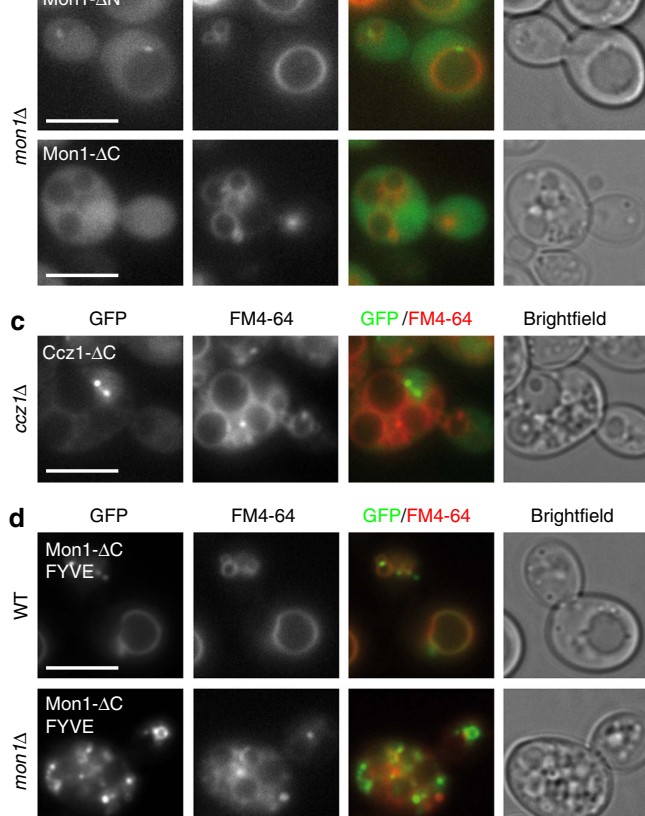

**Figure 1 | Catalytic activity and localization requirements of Mon1–Ccz1.**
(**a**) Nucleotide exchange rates of Ypt7 are plotted as a function of $Ct$MC1 concentrations. The catalytic efficiency of full-length $Ct$MC1 and a truncated $Ct$MC1core complex (Mon1 195–355, Ccz1 1–249) are comparable. Error bars represent s.d. of three independent biological repeats. (**b**) GFP-tagged truncations of $Sc$Mon1 and (**c**) $Sc$Ccz1 are introduced into $mon1Δ$ and $ccz1Δ$ yeast knockout strains. The N-terminus of Mon1 is dispensable, but deletion of the Mon1 and Ccz1 C-termini cause mislocalization and vacuolar fragmentation. (**d**) The C-terminus of Mon1 is replaced by a PI3P-binding FYVE domain and artificially recruited to endosomal and vacuolar membranes as seen in a wild-type background, but Mon1-ΔC-FVYE is not able to complement a $mon1Δ$ strain. Scale bars: 5 μm.

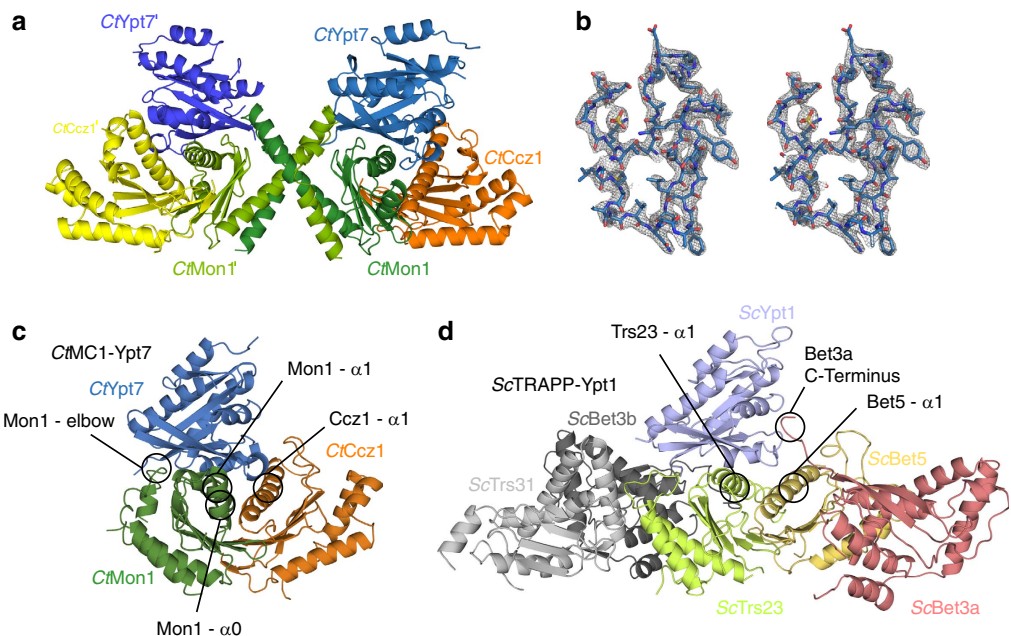

**Figure 2 | Crystal structure of the catalytic *Ct*MC1core complex bound to nucleotide-free *Ct*Ypt7–N125I.** (**a**) The content of the asymmetric unit is shown, comprising two copies of Ypt7 and two Mon1–Ccz1 heterodimers that interact via the domain-swapped helix α3 of Mon1. (**b**) Stereo image of the 2F$_O$–F$_C$ electron density map at 1.8σ contour level of the switch I, switch II and P-loop region of *Ct*Ypt7–N125I bound to MC1. (**c**) Complex structures of MC1 with Ypt7 where the domains' swap was corrected to represent the likely functional unit. (**d**) Complex structure of TRAPP with Ypt1. The different interaction interfaces between GTPase and GEF are marked.

MC1 to endosomal membranes. The Mon1 C-terminus alone did not associate with membranes in a wild-type background, neither did a fusion construct comprising both C-termini of Mon1 and Ccz1 (Supplementary Fig. 2d). Thus, Mon1 and Ccz1 not only cooperate in membrane recruitment by providing additive binding interfaces, the defined arrangement of these modules in the context of the properly assembled MC1 complex seems also necessary.

Previous studies suggested that PI3P binding of MC1 is a major determinant of membrane localization. We wondered whether artificially tethering MC1 to PI3P-positive membranes can rescue its function. The C-terminal domain of Mon1 is predicated to have a strikingly high isoelectric point (*Ct*Mon1: aa 486–665, pI = 9.42), suggesting that its surface is positively charged and thus is prone to interacting with negatively charged lipid head groups like of PI3P. We therefore replaced the C-terminus of Mon1 by the FYVE domain of the EEA1 (early endosomal antigen 1), which has been shown to specifically and robustly interact with PI3P. Although Mon1ΔC-FYVE was efficiently recruited to endosomes and vacuoles, vacuolar morphology was not restored (Fig. 1d). Taken together, we conclude that functionality of MC1 requires its proper localization to specific endosomal microcompartments, which involves both the termini of Mon1 and Ccz1 and their proper arrangement within the complex.

**Crystal structure of the catalytic *Ct*MC1core with Ypt7.** To gain mechanistic insight into MC1 function, we determined the crystal structure of the catalytically active *Ct*MC1core bound to the nucleotide-free *Ct*Ypt7–N125I. The structure was refined to 2.5 Å resolution revealing two *Ct*MC1–Ypt7 complexes per asymmetric unit (Table 1 and Fig. 2a,b). Both complexes are structurally highly similar (r.m.s.d. 0.324 Å over 411 C$_\alpha$ atoms; ref. 30). Thus, the following structural analyses will refer to the complex of *Ct*Ypt7 chain C, which shows better defined

electron density. We used full-length *Ct*Ypt7 for crystallization, but the hypervariable region is not resolved in the electron density map. Mon1 as well as Ccz1 adopt the typical longin domain architecture (Supplementary Fig. 3), with Mon1 harbouring an additional α-helix α0 at the N-terminus (D211 to G220). This helix is conserved in Mon1—including the yeast and human protein—and an integral part of the globular Mon1 fold. Two extensive loops in *Ct*Ccz1 between β-strand β2 and α-helix α1 as well as β5 and α2 are disordered. The latter is flanked by two short additional β-strands. In the asymmetric unit the C-terminal α-helices α3 of both *Ct*Mon1 molecules are swapped such that a dimer of the trimeric *Ct*MC1–Ypt7 complex is formed (Fig. 2a). This likely explains the trimer/hexamer equilibrium observed for MC1core–Ypt7 on gel filtration. Because MC1Δ–Ypt7, which also contains the C-terminus of Mon1, was monomeric, we reason that in the context of full complex, no domain swap occurs and α3 of Mon1 will not interact in *trans*.

We tried to identify a possible hinge where the domain swap might occur. Helix α2 is interrupted by a kink introduced by a proline (P317) and followed by a 15 amino-acid 'elbow loop' that connects α2 and α3, indicating substantial conformational flexibility in this region. It is therefore possible that α2 is bent at position 317 at a different angle in the context of the full complex, which would then allow a different orientation of the elbow loop and helix α3 to interact in *cis*. On the basis of these considerations, we generated a composite model of the likely functional biological unit without domain swap where the elbow loop folds back and α3 completes the longin fold of Mon1 intramolecularly (Fig. 2c and Supplementary Fig. 4a,b).

The heterodimerization of *Ct*Mon1 and *Ct*Ccz1 is mainly mediated by the central β-strands of both proteins, which form a continuous β-sheet, and the α-helices α1, which are oriented on top of the β-sheet alongside another in an antiparallel manner. This α1 surface also represents the main interaction site

**Table 1 | Crystallographic statistics of the *Ct*MC1core–Ypt7 complexes.**

| | *Ct*MC1core–Ypt7 (SAD peak) | *Ct*MC1core–Ypt7 (pdb code: 5LDD) |
|---|---|---|
| *Data collection* | | |
| X-ray source | Beamline P13, EMBL, Hamburg, Germany | |
| Detector | PILATUS 6M | |
| Wavelength (Å) | 0.97958 | 0.97951 |
| Space group | $P2_12_12_1$ | $P2_12_12_1$ |
| Cell dimensions (*a,b,c* Å) | 62.56, 100.57, 207.10 | 70.08, 103.81, 206.67 |
| Resolution (Å) | 48.86-2.89 (3.07-2.89) | 46.38-2.49 (2.65-2.49) |
| Total reflections | 392922 | 346702 |
| Multiplicity | 6.97 | 6.53 |
| Unique reflections | 56335 | 53101 |
| Completeness (%) | 99.7 (98.6) | 99.5 (98.1) |
| $R_{meas}$ (%) | 13.1 (78.5) | 7.6 (90.9) |
| CC 1/2 (ref. 50) | 99.7 (84.8) | 99.9 (91.9) |
| $I/\sigma(I)$ | 11.77 (2.72) | 15.24 (2.62) |
| Mosaicity (°) | 0.153 | 0.134 |
| Wilson *B*-factor (Å²) | 60.89 | 69.32/46.3* |
| | | |
| *Refinement* | | |
| Resolution (Å) | | 46.38-2.50 |
| $R_{work}$, $R_{free}$ | | 0.200, 0.241 |
| Reflections (working, test set) | | 46240, 1839 |
| Completeness for range (%) | | 86.7 |
| r.m.s.d. from ideal | | |
| Bond lenghts (Å) | | 0.004 |
| Bond angles (°) | | 0.620 |
| Total number of atoms | | 7670 |
| Mean *B* value (Å²) | | 71.3 |

Values in parentheses denote the highest resolution shell.
*After anisotropy correction.

with Ypt7 (ref. 31; Supplementary Table 1). The interface of the MC1 complex with Ypt7 is mainly formed by Mon1 ($\sim 960$ Å²) with smaller contributions of Ccz1 ($\sim 450$ Å²). An additional interaction interface is formed by the 'elbow loop' between helices α2 and α3 of Mon1 that interacts with the α3–β5 loop of Ypt7 ($\sim 350$ Å²).

The overall arrangement of the complex resembles the structure of the five-subunit TRAPP GEF complex bound to its substrate Ypt1 (Fig. 2d). The subunits Trs23 and Bet5, homologous to Mon1 and Ccz1, respectively, also form a longin dimer with a central β-sheet and interact with Ypt1 via an interface formed by the α1 helices[32]. Differences are the contribution of a third subunit, Bet3, which inserts its C-terminus into the Ypt1-binding pocket. No equivalent structural feature is present in MC1. Instead, helix α0 and the elbow loop of Mon1 make additional contacts with the GTPase not found for TRAPP. Therefore, the interface between the MC1 longin dimer with 1,760 Å² is significantly larger than between TRAPP Trs23 and Bet5 and Ypt1, which covers only $\sim 1,070$ Å².

**Recognition of Ypt7 over Vps21.** Several GEFs have been shown to have limited selectivity regarding their GTPase substrate. However, since Ypt7 and Vps21 like MC1 are localized to the endosomal compartment, MC1 should discriminate between both proteins. Indeed, *Ct*MC1 did not show measurable activity towards *Ct*Vps21 (Fig. 3a). In a sequence alignment we searched for MC1-interacting residues that are conserved within in the Ypt7 family but differ from the Vps21 family in yeast and *C. thermophilum* (Fig. 3b and Supplementary Fig. 5). We introduced the corresponding mutations in *Ct*Ypt7, which change polarity (Y37R), introduce charge inversion (E47R) or change small to bulky residues (T58K, A76M/G80N) and tested their functionality in the GEF assay (Fig. 3a). The charge inversion

E47R could be tolerated, leading to only a slight decrease in catalytic activity of MC1. For the remaining mutations, however, no MC1-stimulated nucleotide exchange could be detected. Thus, a few key residues render the surface properties of Vps21 incompatible with MC1 interaction and thus guarantee GEF specificity for Ypt7.

**Guanine nucleotide exchange mechanism by the MC1 complex.** We first tested the contributions of the different MC1–Ypt7 interaction interfaces to the efficiency of nucleotide exchange (Fig. 4). On the basis of the homology to TRAPP, the α1 interface has been proposed as important for GEF activity before[17]. We generated three sets of double mutants in the α1 interface of *Ct*MC1, G250W/T254K and G232P/K233D in *Ct*Mon1 and G106W/G110M in *Ct*Ccz1. To test for the requirement of the elbow loop of Mon1, we mutated a conserved SxDxR motif that interacts with Ypt7 (S328W/D330A/R332A) and replaced the entire loop by a GS-linker (S312-[GS]$_6$-E339; Fig. 4a). All mutant complexes expressed well and yielded homogenous protein, but they had no detectable nucleotide exchange activity in the fluorescence GEF assay (Fig. 4b).

For a correlation with functionality *in vivo*, we used *mon1*Δ and *ccz1*Δ yeast deletion strains, which show a vacuolar fragmentation phenotype as a consequence of defective Ypt7 activation in these cells. The expression of GFP–Mon1 or GFP–Ccz1, respectively, rescues the phenotype. We tested mutants of the yeast proteins that are equivalent to the mutations we had introduced in *Ct*MC1 for their ability to complement the knockout (Fig. 4c,d). Consistent with the results from *in vitro* characterization, none of the mutants was able to rescue. All mutants properly localized to endosomal structures in a wild-type background (Supplementary Fig. 6), showing that the observed effects arise from defective interaction of MC1 with

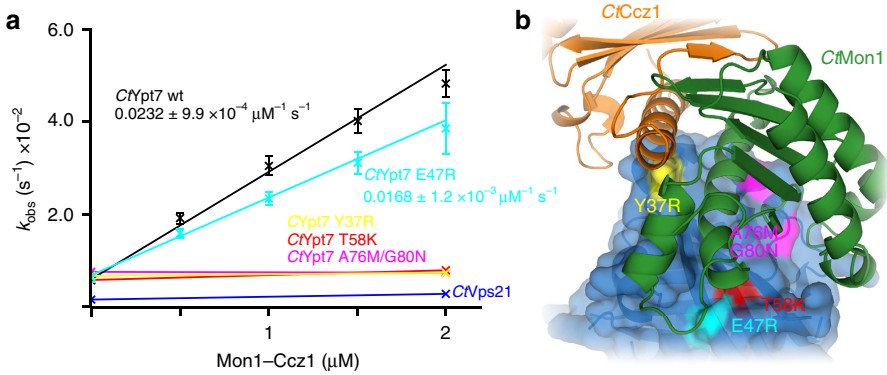

**Figure 3 | Selectivity of Mon1–Ccz1 for Ypt7 over Vps21.** (**a**) For *Ct*Vps21, *Ct*Ypt7 and different mutations, nucleotide exchange rates are plotted as a function of *Ct*MC1core concentration. Error bars represent s.d. of three independent biological repeats. (**b**) MC1-interacting interface of Ypt7 is shown in blue. Positions that are conserved in the Ypt7 family, but not the Vps21 family, are highlighted. Labels show the mutations from Ypt7- to Vps21-specific residues. Colours correspond to the graphs in **b**.

**Figure 4 | Interaction of Mon1–Ccz1 with Ypt7.** (**a**) Surface representation of the *Ct*MC1 longin heterodimer. Mutations introduced in the interaction interface with Ypt7 are labelled. (**b**) Nucleotide exchange rates of Ypt7 are plotted as a function of the concentration of *Ct*MC1core wild-type and different mutations. Error bars represent s.d. of three independent biological repeats. Colours of the graphs correspond to highlighted mutations in **a**. Functionality test of mutations in (**c**) *mon1Δ* and (**d**) *ccz1Δ* yeast knockout strains. The vacuolar fragmentation phenotype is rescued by the introduction of *Sc*Mon1 and *Sc*Ccz1, respectively, but not the mutations corresponding to the GEF-deficient mutants described above. Scale bars: 5 μm.

Ypt7. Importantly, the elbow loop represents a previously unrecognized structural element that is essential for MC1 functionality.

In the *Ct*MC1–Ypt7 complex, the conformation of the Ypt7 P-loop is identical to the nucleotide-bound form and coordinates a sulfate ion from the crystallization condition at the

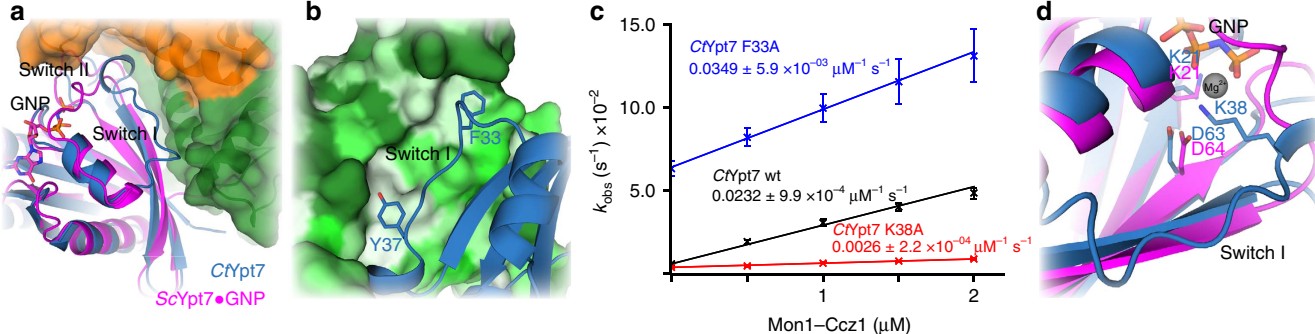

**Figure 5 | Mechanism of nucleotide exchange by Mon1–Ccz1.** (**a**) Superposition of $Sc$Ypt7 bound to the GTP analogue GNP in magenta with MC1-bound $Ct$Ypt7 in blue. The switch I and switch II regions, which undergo major rearrangements, are labelled. (**b**) Interaction of F33 and Y37 from switch I of Ypt7 with two hydrophobic pockets on MC1. The surface of MC1 is coloured according to hydrophobicity from green (hydrophilic) to grey (hydrophobic). (**c**) Nucleotide exchange rates of $Ct$Ypt7 wild type and the mutants F33A and K38A are plotted as a function of the concentration of $Ct$MC1core. Error bars represent s.d. of three independent biological repeats. (**d**) Close-up of the nucleotide pocket from $Sc$Ypt7-GNP and MC1-bound $Ct$Ypt7. The position of switch II D63/64 and P-loop K21 are unaltered. K38 from switch I is inserted into the pocket and clashes with the $Mg^{2+}$ involved in coordinating nucleotides.

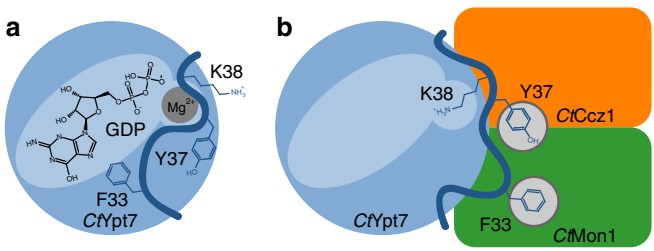

**Figure 6 | Model of the nucleotide exchange mechanism by Mon1–Ccz1.** (**a**) F33 of Ypt7 interacts with the guanosine base to stabilize nucleotide binding. Switch I closes the binding pocket with Y37 and K38 solvent exposed. (**b**) F33 and Y37 are fixed in hydrophobic pockets on Mon1–Ccz1, leading to switch I rearrangement and opening of the nucleotide-binding pocket. In addition, K38 of Ypt7 is inserted into the nucleotide pocket, repelling with its positively charged terminal amine group the $Mg^{2+}$ cofactor ion from the binding site.

position that would be occupied by the β-phosphate of GDP or GTP. The presence of a phosphate or sulfate ion at this position is a common feature of some GEF–GTPase complex crystal structures[33,34]. Interaction with MC1 leads to a dramatic remodelling of the nucleotide-binding pocket of Ypt7 (Fig. 5a). Switch I is moved 18 Å and switch II 8 Å compared with the active conformation observed for $Sc$Ypt7 in complex with the GTP analogue GNP (guanosine 5′-[β,γ-imido]triphosphate)[35]. The entire switch I is ordered and held in place by the interaction of the aromatic switch I residues F33 and Y37 with hydrophobic binding pockets on MC1 (Fig. 5b). The interaction of Y37 with MC1 is essential for GEF function, as nucleotide exchange of $Ct$Ypt7–Y37R was no longer stimulated by $Ct$MC1 (Fig. 3a). F33 is a conserved key residue that was described to stabilize nucleotide binding to Ypt7 via edge-to-face interactions[35]. As expected, the intrinsic nucleotide exchange rate of a $Ct$Ypt7-F33A mutant was strongly elevated by an order of magnitude (Fig. 5c and Supplementary Fig. 7a). The dislocation of this aromatic residue from the binding pocket was reported as part of the nucleotide release mechanism for TRAPP[9] and DENND1 (ref. 6). Surprisingly, addition of $Ct$MC1 could still further stimulate nucleotide exchange of $Ct$Ypt7-F33A with a catalytic efficiency comparable to $Ct$Ypt7 wild-type protein ($k_{cat}/K_M$ $3.5 \times 10^4\,M^{-1}\,s^{-1}$, Fig. 5c and Supplementary Fig. 2c), suggesting a negligible role in the mechanism.

Another striking consequence of the switch I conformation imposed by MC1 is that K38 of Ypt7 is inserted into the nucleotide-binding pocket (Fig. 5d). In the active Ypt7 conformation, this residue is surface-exposed and disordered, but in complex with the GEF, the lysine amine group occupies the position of the $Mg^{2+}$ ion in the nucleotide-bound structures. A lysine at this position of switch I is conserved in the Rab7 family (Supplementary Fig. 7b). We envisage that upon MC1 binding to Ypt7, K38 will push the $Mg^{2+}$ out of the binding pocket and thus destabilize nucleotide binding. To test the importance of this mechanism, we generated a $Ct$Ypt7–K38A mutant. The basal exchange rate of the mutant was only slightly reduced by a factor of two compared to wild-type $Ct$Ypt7 (Supplementary Fig. 7a) and the catalytic efficiency of MC1 for this mutant was strongly reduced by an order of magnitude to $2.6 \times 10^3\,M^{-1}\,s^{-1}$ (Fig. 5c). Thus, reorientation of a Rab switch I lysine into the nucleotide-binding pocket represents a novel mechanism employed by a RabGEF.

## Discussion

The analysis of the entire MC1 complex by electron microscopy established that Mon1 and Ccz1 form a globular heterotetramer with two copies of each protein. The complex shows a double-winged overall architecture with a pseudo-twofold symmetry. Mon1 and Ccz1 assemble into this arrangement via two distinct interfaces. The longin domains in both proteins mediate the formation of heterodimers, which then interact with the C-terminal domain of Ccz1 to form the complete complex. The heterotetrameric arrangement is required for full functionality of MC1 in vivo. We find that different molecular functions of MC1 are carried out by distinct domains within the complex: whereas the C-termini of Mon1 and Ccz1 are needed for proper localization of MC1, the longin heterodimer catalyses nucleotide exchange of Ypt7 - but only jointly in the context of the assembled complex these different activities constitute the entire MC1 functionality. For MC1 it has been shown that membrane binding leads to a dramatic increase in catalytic activity[17]. This is likely the result of the locally increased concentration of enzyme and substrate in two dimensions as seen for the GAP Rasal[36]. Whether orientation effects might play a role as well warrants further investigation.

The activation of Rab GTPases is coupled with their binding to membranes. Therefore, the localization of GEFs is crucial for proper recruitment of the cognate Rab protein. Identity of

membranes in the endomembrane system is in part conveyed by phosphatidylinositides, of which different species are concentrated at different membranes by respective kinases and phosphatases. The membranes of the endosomal system are enriched in PI3P[37]. Previous work had shown that binding to PI3P represents an important factor to recruit MC1 to model membranes[17], providing a potential mechanism for MC1 localization to membranes. We find that PI3P binding alone is not sufficient to guarantee functionality of MC1 *in vivo*. When we artificially tethered the catalytically active MC1core to PI3P-positve membranes, yeast cells still showed vacuolar fragmentation indicative of defective Ypt7 activation and impaired fusion. Likely, MC1 has to be concentrated or confined to specific microcompartments, and additional factors like regulatory proteins are needed to coordinate MC1 recruitment. In addition, proper orientation on the membrane could be required. One important factor might be the Rab5-like Vps21, the upstream Rab GTPase in the endosomal Rab cascade[28]. We speculate that Vps21, potentially along with other yet unidentified factors, needs to interact with MC1 to ensure proper activation.

The enzymatic activity of MC1 is mediated entirely by a core complex consisting of the longin domains of Mon1 and Ccz1 plus an extra N-terminal helix of Mon1. The globular shape of both the longin dimer and complete complex do not allow to fit the crystal structure of the core unambiguously into the EM density of full-length MC1. The comparison of the full complex with MC1core at 17 Å resolution shows that the core, which only represents a third of the total complex, could fit into the EM reconstruction at various positions (Supplementary Fig. 4c). The catalytic efficiency $k_{cat}/K_M$ of $\sim 2.1 \times 10^4 M^{-1} s^{-1}$ we measured for MC1 is comparable to that of other RabGEFs and most similar to TRAPP ($k_{cat}/K_M$ $1.6 \times 10^4 M^{-1} s^{-1}$; ref. 9) or DENND1 ($k_{cat}/K_M$ $2.3 \times 10^4 M^{-1} s^{-1}$; ref. 4). For yeast MC1 a $k_{cat}/K_M$ of $1.6 \times 10^3 M^{-1} s^{-1}$ has been reported[17]. We rationalize the 10-fold difference with the poor stability of the complex from *S. cerevisiae* that likely led to an underestimation of *Sc*MC1 activity.

The utilization of a longin heterodimer by MC1 as GEF module is reminiscent of TRAPP. Longin domains are commonly found as interactors, but the interaction interfaces are hardly conserved[18]. Nevertheless, binding of MC1 to Ypt7 and TRAPP to Ypt1 occurs in an identical orientation to a surface formed jointly by the helices α1 from both longin domains. This argues for a divergent evolutionary relationship between MC1 and TRAPP in contrast to other examples described so far. However, besides the obvious difference in the total number of subunits, MC1 and TRAPP are also distinct in the mechanism by which they mediate nucleotide exchange. To promote nucleotide exchange, TRAPP uses in addition to the Trs23/Bet5 longin dimer the C-terminal tail of a third subunit Bet3 as a wedge that is inserted into the nucleotide-binding pocket of the GTPase. The absence of this feature in MC1 is compensated by a more extensive interaction interface between longin domains and GTPase ($1,760 Å^2$ for MC1 versus $1,070 Å^2$ for TRAPP), which is mediated by two conserved structural features in Mon1: the additional N-terminal helix α0 and the elbow loop with an SxRxD motif.

The interaction with MC1 locks the switch I region of Ypt7 in a conformation incompatible with nucleotide binding. Switch I is entirely structured and well defined in our crystal structure, revealing two complimentary mechanisms that stimulate nucleotide exchange (Fig. 6). First, F33 of Ypt7 is fixed in a hydrophobic pocket on Mon1 and is thus not available for interaction with the guanosine base of GTPase-bound nucleotide. Aromatic residues are conserved at this position

of Rab switch I regions, and their extraction from the nucleotide-binding pocket was proposed to represents a common theme in the mechanism of GEFs[6]. In contrast, we find that MC1 can still promote nucleotide exchange with a catalytic efficiency comparable to wild type, indicating that removal of F33 does not play a key role in the catalytic mechanism of MC1. Unexpectedly, the MC1 stabilized conformation of switch I also leads to insertion of K38 of Ypt7 into the nucleotide pocket, thus occupying with its positively charged terminal amine group the binding site of the $Mg^{2+}$ cofactor ion. This position in switch I is variable within the Rab family, but strictly conserved in the Rab7 subfamily (Supplementary Fig. 7b), which is substrate for MC1. This supports its essential role in the GEF mechanism. Furthermore, the necessity for a lysine in switch I ensures specificity of MC1 in addition to recognition requirements as we observed them for Vps21.

The BLOC3 complex (Hps1–Hps4) has been identified in mammalian cells as a homologous complex to MC1 with different substrate specificity for Rab32/38 instead of Rab7 (ref. 13). Both subunits contain a predicted longin domain in their N terminus, but lack the additional helix α0. It is to be expected that Hps1 and Hps4 also form a catalytic longin dimer. Interestingly, both Rab32 and Rab38 contain an arginine residue at the equivalent position to K38 in Ypt7 (Supplementary Fig. 7b). Thus, BLOC3 might utilize the same mechanism as MC1 to catalyse nucleotide exchange.

## Methods

**Cloning of *Ct*MC1 and Rab GTPases.** To identify the heterodimeric MC1 GEF complex in the thermophilic fungus *C. thermophilum*, DELTA-BAST searches (www.ncbi.nlm.nih.gov/BLAST/) were performed using the protein sequences of the *S. cerevisiae* MC1 complex (*Sc*Mon1: UniProt entry P53129, *Sc*Ccz1: UniProt entry P38273) as queries. Homologues of Mon1 and Ccz1 could be found with sequence identities of 29% (NCBI accession code XP_006697030, E value: 6e$^{-53}$) and 15% (XP_006695440, E value: 1e$^{-6}$), respectively. Constructs of *Ct*Mon1 and *Ct*Ccz1 were amplified from codon-optimized synthetic genes (GenScript, Supplementary Table 2). PCR products (for a detailed primer list see Supplementary Table 3) were subcloned into modified expression vectors pCDF-6P and pET-28a-HS, yielding N-terminally tagged GST-*Ct*Mon1 and His$_6$-SUMO-*Ct*Ccz1 fusion proteins, respectively. *Ct*Ypt7 (XP_006696898) as well as *Ct*Vps21 (XP_006697636) were amplified from *C. thermophilum* cDNA (courtesy E. Hurt) and subsequently subcloned into the modified expression vectors pCDF-6P and pQLinkG, respectively. The latter encodes an N terminally glutathione *S*-transferase (GST) fusion protein with an additional TEV cleavage site. The plasmid pQLinkG-*Ct*Ypt7 was used for co-expression of the trimeric *Ct*MC1–Ypt7 complex. Mutants of *Ct*Mon1 and *Ct*Ccz1 as well as *Ct*Ypt7 were generated with the Q5 Site-Directed Mutagenesis Kit (NEB) and subsequently verified by sequencing (Seqlab).

**Co-expression and protein purification of the *Ct*MC1 complex.** *Escherichia coli* Rosetta (DE3) cells (Novagen) were transformed by electroporation with both plasmids pCDF-6P-*Ct*Mon1 and pET-28a-HS-*Ct*Ccz1. Cells were grown in LB medium to OD$_{600}$ 0.6. After cold shock, co-expression was triggered by 0.25 mM isopropyl-β-D-thiogalactoside at 16 °C and cells were harvested after 18 h. Cell pellets were resuspended in buffer I (50 mM NaH$_2$PO$_4$, 500 mM NaCl, 5% glycerol, 2 mM dithiothreitol (DTT), pH 7.5) and cell disruption was performed in the presence of lysozyme, DNase I and protease inhibitors (Pierce Protease Inhibitor Tablets, EDTA-free, Thermo Fisher Scientific) with a French Press. After centrifugation (39,191*g*, 30 min, 4 °C), the supernatant was applied on a self-packed glutathione column (Pierce Glutathione Superflow Agarose, Thermo Fisher Scientific). Proteolytic cleavage of expression tags was achieved by overnight incubation at 4 °C with PreScission (GST-*Ct*Mon1) and SUMO protease (His$_6$-SUMO-*Ct*Ccz1), respectively. After elution, the *Ct*MC1 complex was concentrated with an Amicon Ultra concentrator (Merck Millipore). As a final polishing step, size exclusion chromatography was performed to separate the heterodimeric GEF complex from proteases. For EM studies, *Ct*MC1 in buffer II (10 mM HEPES, 200 mM NaCl, 0.5 mM TCEP, pH 7.3) was used, whereas the GEF activity assay was performed in buffer III (10 mM HEPES, 200 mM NaCl, 1 mM MgCl$_2$, 5% glycerol, 0.5 mM TCEP, pH 7.3). The *Ct*MC1core complex was purified according to the protocol, but elution was performed with 20 mM GSH in buffer I supplemented with 10 mM DTT to retain both expression tags.

To obtain the trimeric *Ct*MC1–Ypt7 complex *Escherichia coli* Rosetta (DE3) cells were transformed by electroporation with latter plasmids as well as pQLinkG-*Ct*Ypt7–N125I. Triple co-expression and cell disruption was performed

as described for the CtMC1 complex. The protein purification protocol was modified such that after 2 h of PreScission and SUMO protease treatment GSH beads were washed with buffer I to remove all unbound proteins, namely proteases and excess CtMC1. GSH beads with remaining trimeric CtMC1–Ypt7–N125I complex and CtYpt7–N125I alone were incubated with TEV protease at 4 °C overnight and subsequently eluted. For protein crystallization, gel filtration was performed in buffer IV (25 mM HEPES, 300 mM NaCl, 5% glycerol, 1 mM TCEP, pH 7.5) to separate the trimeric CtMC1–Ypt7 complex from TEV protease as well as CtYpt7–N125I and yielded >95% pure protein.

For the purification of CtYpt7 and CtVps21 E. coli Rosetta (DE3) cells were chemically transformed with the expression plasmid pCDF-6P-CtYpt7 and pCDF-6P-CtVps21, respectively. After inoculation, cells were grown in terrific broth medium at 25 °C for 24 h and gene expression was controlled by autoinduction. Both Rab GTPases were purified according to the protocol of the CtMC1 complex with the exception that only PreScission protease was used for proteolytic cleavage of the expression tag. Gel filtration was performed in buffer III.

Expression and purification of all mutants and their complexes were performed as described above.

**Electron microscopy.** Four microlitres of each sample were adsorbed for 2 min at 25 °C on glow-discharged carbon-coated copper grids. The grids were washed twice with the appropriate purification buffer and negatively stained with 0.75% uranyl formate. Samples were imaged on a JEOL JEM-1400 equipped with a LaB$_6$ cathode operated at 120 kV. Images were recorded on a 4k × 4k charge-coupled device camera F416 (TVIPS) using minimal dose conditions. After manual selection of the single particles using EMAN2 (ref. 38), reference-free and reference-based alignment as well as K-means and ISAC classifications were performed using SPARX[39]. In total, 22,165 particles were aligned and classified into 100 classes. These ISAC classes provided the templates for ab initio 3D structure determination with sxviper (SPARX). The initial model was subsequently refined using the raw single particles. The resolution of the final reconstruction, estimated using a Fourier shell correlation criterion of 0.5, was calculated to be ~17 Å (Supplementary Fig. 1e).

**GEF activity assay.** Purified CtYpt7 and CtVps21 were loaded with MANT–GDP (Jena Bioscience) in the presence of 20 mM EDTA and 1.5 molar excess of fluorescent nucleotide at 4 °C overnight. Loading reaction was quenched by the addition of MgCl$_2$ to 25 mM and the resulting Rab GTPase–MANT–GDP complex purified via size exclusion chromatography in buffer III. For the GEF activity assay, 2.0 μM Rab GTPase–MANT–GDP complex were pre-incubated with 2.0, 1.5, 1.0, 0.5 and 0 μM of respective CtMC1 complex. After baseline stabilization, the nucleotide exchange reaction was triggered by the addition of 0.1 mM GTP. Substitution of MANT–GDP for GTP upon GEF activity was monitored by the decrease in fluorescence emission at $\lambda_{em}$ 450 nm ($\lambda_{ex}$ 354 nm) in intervals of 60 s at 25 °C. Data were fitted against a first-order exponential decay ($y = y_0 + A^* \exp(-x/t)$) and $k_{obs}$ (s$^{-1}$) was determined by $k_{obs} = 1/t$. Subsequently, $k_{obs}$ was plotted against the CtMC1 concentration and $k_{cat}/K_M$ (M$^{-1}$ s$^{-1}$) was determined as the slope of the linear fit $y = A^* x + B$. The measured $k_{obs}$ of the intrinsic nucleotide exchange was used as data point for 0 μM CtMC1 concentration.

**Protein crystallization and structure determination.** An initial crystallization screening was performed with commercially available crystallization screens (Molecular Dimensions) and a Gryphon robot system (Art Robbins Instruments) in a 96-well format. First crystals were obtained at 12 °C and a protein concentration of 6.3 mg ml$^{-1}$ after several weeks in a crystallization condition containing 0.2 M sodium chloride, 0.1 M HEPES sodium salt, pH 6.5 and 10% PEG 4000. Subsequent microseed matrix screening[40] identified a reliable crystallization condition containing 0.15 M ammonium sulfate, 0.1 M MES, pH 6.0 and 15% PEG 4000. Optimization was done in a 24-well format using the hanging-drop vapour diffusion method at 12 °C and a protein concentration of 7.3 mg ml$^{-1}$. Native protein crystals for data collection were obtained after a few days using the streak seeding technique[41] in latter crystallization condition but with 18% PEG 4000 and supplemented with 25% glycerol. Native crystals were directly flash-cooled in liquid nitrogen. To obtain phase information, selenomethionine-substituted protein was produced[42], purified and crystallized in the native crystallization condition supplemented with 15% glycerol using the streak seeding technique with seeds from native crystals. Selenium-derivative crystals were flash-cooled in liquid nitrogen in latter condition with 17% PEG 4000 and with 25% glycerol as cryoprotectant.

Native as well as anomalous X-ray data were collected from single crystals at 100 K at beamline P13, EMBL Hamburg, Germany. Diffraction data were processed using XDSAPP[43]. Initial phases were determined by single-wavelength anomalous dispersion at the selenium peak energy using phenix.autosol[44,45], followed by density modification and automated model building using phenix.autobuild[44,46]. Since the diffraction pattern showed strong anisotropy, processed native data were analysed through the UCLA Diffraction Anisotropy Server[47] (http://services.mbi.ucla.edu/anisoscale) and ellipsoidally truncated to 2.9, 2.5 and 2.5 Å along a*, b* and c*, respectively. After anisotropic scaling, a

negative isotropic B-factor of −22.01 Å$^2$ was applied to correct data for further refinement. Iterative cycles of model building in COOT[48] and refinement at 2.5 Å using phenix.refine[44,49] led to a final model with $R_{factors}$ of $R_{work}$ 20.0% and $R_{free}$ 24.1%. Data collection and refinement statistics are summarized in Table 1. Coordinates and structure factors have been deposited in the Protein Data Bank (pdb code: 5LDD). Protein interfaces and interactions were calculated using PISA[31] (http://www.ebi.ac.uk/pdbe/pisa/). PYMOL was used for graphical analysis and visualization[30].

**Fluorescence microscopy.** Microscopic analyses of yeast cells[17] were performed as described in Cabrera et al.[17] Cells were grown in YPD overnight, diluted to OD$_{600}$ of 0.25 in the morning and grown until an OD$_{600}$ of ~1. Cells were collected by centrifugation (5,000g, 3 min, 20 °C) and washed in synthetic media. For staining of the vacuole by FM4-64, cells were incubated in synthetic media containing 30 μM FM4-64 for 30 min at 30 °C, washed twice in fresh media and incubated another 60 min in media without dye. Images were acquired directly afterwards using a Delta Vision Elite (GE Heathcare) equipped with an inverted microscope (model IX-71; Olympus), an UAPON × 100 (1.49 numerical aperture (NA)) oil immersion or PLAPON × 60 (1.42 NA) oil immersion objective, an InsightSSI light source (Applied Precision) and an sCMOS camera (PCO). Data were processed using ImageJ 2.0.0.

**Data availability.** The coordinates and structure factors of the CtMC1core–Ypt7 complex have been deposited in the Protein Data Bank under the accession codes 5LDD. All additional experimental data are available from the corresponding author on reasonable request. The UniProt accession codes P53129 and P38273 and the NCBI accession codes XP_006697030, XP_006695440, XP_006696898 and XP_006697636 were used in this study.

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

## Acknowledgements

We acknowledge the Helmholtz-Zentrum Berlin for provision of synchrotron radiation beamtime at beamline MX14.1 & MX14.3 of BESSY II and would like to thank Uwe Müller for assistance. We are grateful to Johanna Kallio, Michele Cianci and the staff of the EMBL beamline P13 at the PETRA III synchrotron for excellent support during data collection and Siegfried Engelbrecht-Vandré for helpful discussions. We are grateful to Eric Herrmann for technical assistance. L.L. was supported in part by an incentive award of the Department of Biology/Chemistry in Osnabrück. This study was supported by grants from DFG to D.K. (KU2531/2-1, SFB 944-P17), C.U. (SFB944-P11) and S.R. (RA 1781/2-3).

## Author contributions

S.K., C.U. and D.K. conceived the study and designed the biochemical and cell biology experiments; S.K. purified and crystallized proteins, performed kinetic analysis and solved the structure with help from S.S. and D.K.; A.K. and S.R. designed, performed and analysed the EM experiments; L.L. performed cell biology experiments; S.K. and D.K. wrote the manuscript; all authors contributed to discussion and approved the final manuscript.

## Additional information

**Competing financial interests:** The authors declare no competing financial interests.

