## [Peer review file · Nature Communications]

Reviewers' comments:

Reviewer #1 (Remarks to the Author):

The manuscript „Architecture and mechanism of the Ypt7/Rab7 guanine nucleotide exchange factor complex Mon1-Ccz1" by Kiontke et al. reports on the long-awaited structural mechanism of the activation of Rab7-family proteins. The authors present the crystal structure and biochemical characterization of the heterodimeric guanine nucleotide exchange factor (GEF) Mon1-Ccz1 (MC1).

Rab proteins are essential regulators of intracellular vesicular trafficking. They require activation by GEFs in order to become active. The enzymatic activation is mediated through the catalytic replacement of guanosine diphosphate (GDP) by guanosine triphosphate (GTP), resulting in the generation of active, GTP-bound Rab proteins. However, the process of Rab activation is understood structurally only for a number of the 60 human Rab proteins, since Rab GEFs are generally not conserved in sequence or structure. Consequently, the corresponding Rab-GEF structures need to be determined individually for each Rab subfamily.

The Rab protein Rab7/Ypt7 was among the first discovered Rab proteins. Due to its importance in the processing of endocytic material, it has received great scientific attention and represents one of the best understood Rab proteins on a cellular level. The mechanism of its activation, however, could not be addressed previously since the GEF MC1 was elusive for many years and even after identification failed to be structurally characterized.

The group of Daniel Kuemmel has now made a major breakthrough in the structural understanding of the activation of Rab7/Ypt7 by MC1. The researchers successfully determined the Ypt7-MC1 complex structure. By using mutagenesis studies in combination with preliminary kinetic investigations they were able to deduce first principles of the activation process and its structural determinants. Also, the group has used cell biology approaches to understand the relevance of point mutations in a cellular context.

The results obtained by the authors clearly warrant publication in nature communications. The authors should however improve their manuscript and figures for clarity. In the following I will provide my suggestions on this matter.

Specific comments:

- please provide a brief rationale why MC1 from *C. thermophilum* has been used for the structural characterization. The uninformed reader may not understand why this organism has been chosen as a source of the genetic material.

- Figure 1: The data are irrelevant for the manuscript and should be moved to the supplement. It appears that no relevant information is deduced from the EM-data. A rationale of the biological significance of the EM-data is missing and no interpretation is given in the result or discussion section. If the authors wish to present these data in the main text, they must provide more

explanations and discussions of them.

- The authors give the following statement on page 4 (top): "The core complex still bound Ypt7 but showed a trimer/hexamer equilibrium that likely is an artefact caused by the truncated Mon1 construct and does not reflect a functional interaction mode (see below)." Please provide an experimental rationale why this trimer/hexamer is thought to be an artifact (or point at the section in the manuscript where a rationale is given).

- The manuscript is lacking the experimental details for the calculation of the catalytic efficiencies (k_{cat}/K_m) of the GEF. Please provide a detailed procedure.

- Also the observed rate constants (k_{obs}) appear to be calculated incorrectly. The authors state on page 11 in their experimental procedures "GEF activity assays" that they have fitted their fluorescence data to a single exponential. Subsequently, they have calculated k_{obs} via the formula $k_{obs} = \ln 2 / (t_{1/2})$. However, the fit to a single exponential of the type $e^{-x/t}$ is not giving the half-life $t_{1/2}$ of the reaction. In this type of the equation, the relationship between t and k_{obs} is: $k_{obs} = 1/t$. Since the exponential fit function is not presented in the method section and no reference is provided, the correctness of the data interpretation cannot be evaluated here. I would like to ask the authors to provide more details and references.

- There must not be any indication of "data not shown" in the manuscript (page 6, end of second paragraph). Data can be provided in the supplement.

- page 5, section "crystal structure of the catalytic...": The origin, relevance, and interpretation of P317 is not clear to me. Please provide more information and a more comprehensible interpretation and indication of significance.

- page 6, Heading "Guanosine nucleotide exchange...": Must read "Guanine ...". Also, "...in the fluorescent GEF assay (Fig. 5b)" should read "in the fluorescence GEF assay...".

- page 6, bottom: The authors mention that F33 of Ypt7 is important for nucleotide binding. Please provide a rationale, why it is involved in nucleotide binding and how this is achieved on a molecular level. Additionally, the authors show that the F33A substitution is still an excellent substrate to MC1. However, they state that this is surprising. Please provide a rationale, why this is surprising in the view of the authors.

- The units and numbers in the figures should correspond in their format to the main text. E.g. Figure 2a: Exponential representations should be given not with "e-2" et cetera.

- The authors should refrain showing surface representations together with cartoon representations. The authors should delete all surface representation, perhaps with the exception of figure 6 a, b.

- The authors should present a numbering and designation of the secondary structure elements of their complex structure (could be part of the supplement).

- Figures 4a, 5b: It appears as if the authors have calculated their k_{cat}/K_m values in some instances from experiments having only 2 data points. This is inappropriate and the authors must provide more data. Otherwise, a statement on the significance of the E47R mutant on the GEF activity (figure 4a) is not possible.
- The authors should provide a list or depiction of the interactions between MC1 and Ypt7. There they should highlight the most important protein-protein interactions.
- Supplementary figure 3 would be better suited for the main text than figure 3a.
- The authors should also provide a schematic figure that summarizes the nucleotide exchange reaction catalyzed by MC1.

Reviewer #2 (Remarks to the Author):

This is an important and concise manuscript describing the structure of the Mon-Ccz1 complex (MC1) bound to Ypt7. It is mainly focussed on the architecture of the GEF core and does not really address other issues relating to GEF targeting to membranes. Although Ypt7/Rab7 has been known for many years to be the Rab GTPase required for endosomal maturation and fusion at the vacuole/lysosome, details of its regulation have been lacking. This structure is of importance since it provides a molecular perspective for how the GEF discriminates Ypt7 over Vps21, the Rab5 related GTPase located to early endosomal membranes. Overall there is enough high quality work of relatively broad appeal to justify publication in Nature Comms. However, I have a few comments that the authors should address prior to publication.

Although the title reflects well the content of the paper, it fails to mention where in the cell the Mon1-Ccz1 complex acts. Perhaps late endosomes/lysosomes could be mentioned in the title.

The authors present a 3D reconstruction of the MtMC1full complex. While the images are beautiful, the EM data does not seem to add much to the study since it is not sufficiently linked to the X-ray structure for the MC1core and other biochemical and cell biological data that follows. I also have a few questions about the way this was produced. The sample is heterogeneous (supplementary figure 1), however the materials and methods do not describe how this heterogeneity was addressed. The dataset comprises of 22165 particles which were classified into 100 classes (on average about 200 particles per class). Would the 3D reconstruction have benefited from the bigger number of classes with fewer particles in each class? The classes in fig 1b show side views or intermediate between side and top views, raw particles on fig 1a also predominantly reveal the side views of the complex suggesting a preferential orientation of the complex on the grid. There is no figure showing top views and no description how preferential orientation of the complex on the grid

was handled. The paper would benefit from a more detailed description of the EM work as well as supplemental figures showing classes and the 3D model in all 3 orthogonal views. The resolution of the final reconstruction based on a Fourier shell correlation criteria of 0.5 was estimated to be about 17Å. However the authors state that it was not possible to fit the crystal structure to the EM envelope. Isn't this a concern? There is no secondary validation of the EM map. Another low resolution method, SAXS for example, might be used to validate the EM envelope. More importantly, there is no description in the manuscript of how the EM reconstruction benefits understanding of the biology of the complex. From my point of view, the manuscript has enough data without the EM, however if EM data is included in the manuscript, it needs to be better explained and more carefully integrated into the story.

Some additional description and supplementary data are needed when MC1core is introduced. It would be beneficial to show the alignments from the different organisms as well as to illustrate the secondary structure elements. When core binding to Ypt7 is described there is no link to supplementary Figure 1d.

The authors state that MtMC1full is a dimer based on the gel filtration profile, however the precise functional relevance isn't clear and this is not definitively established. Taking into account that MtMC1full has an elongated shape from the EM data, gel filtration cannot provide an accurate molecular weight of the complex. It would therefore be useful to add SEC-MALLS data including the molar mass distribution for all complexes used in the study if the authors have this.

With regards to the crystallography described in the manuscript, both datasets are in the same space group and cell dimensions, in essence the same crystal form (including anisotropy) unless I have misunderstood something. However, the high-resolution data set has about 3000 fewer unique reflections compared to the low resolution (SAD peak) dataset and both data sets have 99% completeness. Is this correct? Since the Rmeas values are relatively high, please add the CC1/2 values for both datasets.

Reviewer #3 (Remarks to the Author):

The authors have determined the structure of the Ypt7/Rab7 guanine nucleotide exchange factor complex Mon1-Ccz1 (MC1 complex) at ~2.5Å resolution. The structure provides clues to features that allow discrimination of the Rab7-like Ypt7 over the Rab5-like Vps21 which are both located on the same membrane. The MC1 complex plays an essential role in a Rab cascade that defines the endolysosomal system and orchestrates the Rab switch between Rab5 and Rab7. The heterodimeric MC1 complex from *S. cerevisiae* has been successfully used in biochemical studies, but low expression levels, aggregation, and instability of the recombinant proteins have hampered structural studies. Authors have used Mon1 and Ccz1 from *Chaetomium thermophilum*, which show an analogous domain architecture compared to the *S. cerevisiae* MC1 complex. The full-length CtMC1 has been obtained and a variety of mutants produced to combine with the insight gained from the crystallographic structure of the catalytic CtMC1core in complex with Ypt7 to provide a significant advance in our understanding of the mechanism for this class of RabGEFs. The paper is recommended for publication. Some minor points need addressing:

1. Authors state "The globular shape of both the longin dimer and complete complex do not allow to

fit the crystal structure of the core unambiguously into the EM density of full-length MC1" but offer no explanation,

2. Crystallographic data has somewhat limited completeness (86%), Wilson B is high etc. These should be commented upon (ideally a more complete data should be obtained). If not immediately possible then atleast electron density for some regions of the structure should be shown.

Response to reviewers' comments

Reviewer #1

The manuscript „Architecture and mechanism of the Ypt7/Rab7 guanine nucleotide exchange factor complex Mon1-Ccz1" by Kiontke et al. reports on the long-awaited structural mechanism of the activation of Rab7-family proteins. The authors present the crystal structure and biochemical characterization of the heterodimeric guanine nucleotide exchange factor (GEF) Mon1-Ccz1 (MC1).

Rab proteins are essential regulators of intracellular vesicular trafficking. They require activation by GEFs in order to become active. The enzymatic activation is mediated through the catalytic replacement of guanosine diphosphate (GDP) by guanosine triphosphate (GTP), resulting in the generation of active, GTP-bound Rab proteins. However, the process of Rab activation is understood structurally only for a number of the 60 human Rab proteins, since Rab GEFs are generally not conserved in sequence or structure. Consequently, the corresponding Rab-GEF structures need to be determined individually for each Rab subfamily.

The Rab protein Rab7/Ypt7 was among the first discovered Rab proteins. Due to its importance in the processing of endocytic material, it has received great scientific attention and represents one of the best understood Rab proteins on a cellular level. The mechanism of its activation, however, could not be addressed previously since the GEF MC1 was elusive for many years and even after identification failed to be structurally characterized.

The group of Daniel Kuemmel has now made a major breakthrough in the structural understanding of the activation of Rab7/Ypt7 by MC1. The researchers successfully determined the Ypt7-MC1 complex structure. By using mutagenesis studies in combination with preliminary kinetic investigations they were able to deduce first principles of the activation process and its structural determinants. Also, the group has used cell biology approaches to understand the relevance of point mutations in a cellular context.

The results obtained by the authors clearly warrant publication in nature communications. The authors should however improve their manuscript and figures for clarity. In the following I will provide my suggestions on this matter.

Specific comments:

- please provide a brief rationale why MC1 from *C. thermophilum* has been used for the structural characterization. The uninformed reader may not understand why this organism has been chosen as a source of the genetic material.

We now state in the beginning of the results section: "Proteins from the thermophilic fungus *Chaetomium thermophilum* have been shown to be better suited for structural studies than homologs from other species."

- Figure 1: The data are irrelevant for the manuscript and should be moved to the supplement. It appears that no relevant information is deduced from the EM-data. A rationale of the biological significance of the EM-data is missing and no interpretation is given in the result or discussion section. If the authors wish to present these data in the main text, they must provide more explanations and discussions of them.

The EM structure of MC1 clearly confirms, as suggested by gel filtration, that the complex represents a heterotetrameric assembly, thereby defining the principal architecture of the complex. It also represents the first overall structural description of MC1, and we believe the data should therefore remain in the main part of the manuscript. We thus have included additional description and discussion of the EM data.

- The authors give the following statement on page 4 (top): "The core complex still bound Ypt7 but showed a trimer/hexamer equilibrium that likely is an artefact caused by the truncated Mon1 construct and does not reflect a functional interaction mode (see below)." Please provide an experimental rationale why this trimer/hexamer is thought to be an artifact (or point at the section in the manuscript where a rationale is given).

We now explain: "Because MC1 Δ •Ypt7, which additionally contains the C-terminus of Mon1, elutes as a trimer, we conclude that hexamer formation of MC1core•Ypt7 is an artifact from the truncation of Mon1 that does not reflect a functional interaction mode and does not occur in the context of full complex."

- The manuscript is lacking the experimental details for the calculation of the catalytic efficiencies (k_{cat}/K_m) of the GEF. Please provide a detailed procedure. Also the observed rate constants (k_{obs}) appear to be calculated incorrectly. The authors state on page 11 in their experimental procedures "GEF activity assays" that they have fitted their fluorescence data to a single exponential. Subsequently, they have calculated k_{obs} via the formula $k_{obs} = \ln 2 / (t_{1/2})$. However, the fit to a single exponential of the type $e^{-x/t}$ is not giving the half-life $t_{1/2}$ of the reaction. In this type of the equation, the relationship between t and k_{obs} is: $k_{obs} = 1/t$. Since the exponential fit function is not presented in the method section and no reference is provided, the correctness of the data interpretation cannot be evaluated here. I would like to ask the authors to provide more details and references.

The calculation of k_{obs} was described incorrectly, but performed correctly in the way outlined by the reviewer. The experimental procedures have been modified accordingly and additional information was included so that we now provide a detailed account on the calculation of catalytic efficiencies: "Data was fitted against a first order exponential decay ($y = y_0 + A \cdot \exp(-x/t)$) and k_{obs} (s^{-1}) was determined by $k_{obs} = 1/t$. Subsequently, k_{obs} was plotted against the CtMC1 concentration and k_{cat}/K_M ($M^{-1}s^{-1}$) was determined as the slope of the linear fit $y = A \cdot x + B$. The measured k_{obs} of the intrinsic nucleotide exchange was used as data point for 0 μM CtMC1 concentration."

- There must not be any indication of "data not shown" in the manuscript (page 6, end of second paragraph). Data can be provided in the supplement.

We removed this reference.

- page 5, section "crystal structure of the catalytic...": The origin, relevance, and interpretation of P317 is not clear to me. Please provide more information and a more comprehensible interpretation and indication of significance.

We extended the section and state now: "We tried to identify a possible hinge where the domain swap might occur. Helix α_2 is interrupted by a kink introduced by a proline (P317) and followed by 15 amino acid "elbow loop" that connects α_2 and α_3 , indicating substantial conformational flexibility in this region. It is therefore possible that α_2 is bent at position 317 at a different angle in the context of the full complex, which would then allow a different orientation of the elbow loop and helix α_3 to interact in cis. Based on these considerations, we generated a composite model of the likely functional biological unit without domain swap where the elbow loop folds back and α_3 completes the longin fold of Mon1 intramolecularly (Fig. 3c, Supplementary Fig 4 a,b)."

- page 6, Heading "Guanosine nucleotide exchange...": Must read "Guanine ...". Also, "...in the fluorescent GEF assay (Fig. 5b)" should read "in the fluorescence GEF assay...".

We corrected the mistakes.

- page 6, bottom: The authors mention that F33 of Ypt7 is important for nucleotide binding. Please provide a rationale, why it is involved in nucleotide binding and how this is achieved on a molecular level. Additionally, the authors show that the F33A substitution is still an excellent substrate to MC1.

However, they state that this is surprising. Please provide a rationale, why this is surprising in the view of the authors.

We now elaborate: "F33 is a conserved key residue that was described to stabilize nucleotide binding to Ypt7 via edge-to-face interactions. As expected, the intrinsic nucleotide exchange rate of a CtYpt7-F33A mutant was strongly elevated by an order of magnitude. The dislocation of this aromatic residue from the binding pocket was reported as part of the nucleotide release mechanism for TRAPP and DENND1. Surprisingly, addition of CtMC1 could still further stimulate nucleotide exchange of CtYpt7-F33A with a catalytic efficiency comparable to CtYpt7 wild-type protein, suggesting a negligible role in the mechanism."

- The units and numbers in the figures should correspond in their format to the main text. E.g. Figure 2a: Exponential representations should be given not with "e-2" et cetera.

The units and values are now given in an identical format throughout the manuscript

- The authors should refrain showing surface representations together with cartoon representations. The authors should delete all surface representation, perhaps with the exception of figure 6 a,b.

The use of combined cartoon and surface representation conveys essential information when the location of surface mutations or interfaces are to be presented, as in Fig. 4b, 5a, 6a and 6b. We, however, have removed surfaces from Fig 3 according to the reviewer's request.

- The authors should present a numbering and designation of the secondary structure elements of their complex structure (could be part of the supplement).

We have added sequence alignments of the crystallized parts of Mon1, Ccz1 and Ypt7 from *C. thermophilum* and *S. cerevisiae* and indicated the secondary structure elements as Supplementary Figure 3.

- Figures 4a, 5b: It appears as if the authors have calculated their k_{cat}/K_m values in some instances from experiments having only 2 data points. This is inappropriate and the authors must provide more data. Otherwise, a statement on the significance of the E47R mutant on the GEF activity (figure 4a) is not possible.

We have added more data points for the mutant E47R (Fig 4a), but also F33A and K38A (Fig 6c), which confirm the previously reported results.

- The authors should provide a list or depiction of the interactions between MC1 and Ypt7. There they should highlight the most important protein-protein interactions.

We have included a Supplementary table that lists the interacting residues of Ypt7 and the corresponding residues in Mon1 and Ccz1.

- Supplementary figure 3 would be better suited for the main text than figure 3a.

Supplementary Fig 3 b,c was moved in the main text and Fig 3 b,c to supplement instead.

- The authors should also provide a schematic figure that summarizes the nucleotide exchange reaction catalyzed by MC1.

A schematic model of the MC1 enzymatic mechanism is now included as Fig 7.

Reviewer #2 (Remarks to the Author):

This is an important and concise manuscript describing the structure of the Mon-Ccz1 complex (MC1) bound to Ypt7. It is mainly focussed on the architecture of the GEF core and does not really address other issues relating to GEF targeting to membranes. Although Ypt7/Rab7 has been known for many years to be the Rab GTPase required for endosomal maturation and fusion at the vacuole/lysosome, details of its regulation have been lacking. This structure is of importance since it provides a molecular perspective for how the GEF discriminates Ypt7 over Vps21, the Rab5 related GTPase located to early endosomal membranes. Overall there is enough high quality work of relatively broad appeal to justify publication in Nature Comms. However, I have a few comments that the authors should address prior to publication.

Although the title reflects well the content of the paper, it fails to mention where in the cell the Mon1-Ccz1 complex acts. Perhaps late endosomes/lysosomes could be mentioned in the title.

We changed the title to: Architecture and mechanism of the late endosomal Ypt7/Rab7 guanine nucleotide exchange factor complex Mon1-Ccz1

The authors present a 3D reconstruction of the MtMC1full complex. While the images are beautiful, the EM data does not seem to add much to the study since it is not sufficiently linked to the X-ray structure for the MC1core and other biochemical and cell biological data that follows.

We have included additional description and discussion of the EM data to better represent its significance for defining the overall architecture of the full complex.

I also have a few questions about the way this was produced. The sample is heterogeneous (supplementary figure 1), however the materials and methods do not describe how this heterogeneity was addressed.

The sample is not heterogeneous at all. Only one well-separated peak was taken after size exclusion chromatography as shown in Supplementary Fig. 1. The single particles and the corresponding class averages shown in Fig. 1/Supplementary Fig 1 clearly show that MC1 is neither heterogeneous nor very flexible. The class averages in Figure 1 represent the different orientations of the same complex.

The dataset comprises of 22165 particles which were classified into 100 classes (on average about 200 particles per class). Would the 3D reconstruction have benefited from the bigger number of classes with fewer particles in each class?

For the 3D reconstruction only the first reference is created using the class averages. Later on, the original particles are used (see Methods). Therefore, the number of classes is not relevant. In general, one chooses only more classes if the class averages do not show clear features and have smeared-out borders. This is not the case for the MC1 data set.

The classes in fig 1b show side views or intermediate between side and top views, raw particles on fig 1a also predominantly reveal the side views of the complex suggesting a preferential orientation of the complex on the grid. There is no figure showing top views and no description how preferential orientation of the complex on the grid was handled.

An elongated complex naturally does not lie head-on on the carbon layer. However, we have different side views and intermediates between side and top views, i.e. we do not have preferred orientation. This is an ideal case for performing a 3D reconstruction and a 100% top view is not needed. We have added this information in the revised manuscript and also included top views of the 3D reconstruction.

The paper would benefit from a more detailed description of the EM work as well as supplemental figures showing classes and the 3D model in all 3 orthogonal views.

We have already given a detailed description of our work in the Methods section where all necessary steps are explained. We have added now all 100 class averages in Fig 1 and also present all orthogonal views of the 3D reconstruction.

The resolution of the final reconstruction based on a Fourier shell correlation criteria of 0.5 was estimated to be about 17Å. However the authors state that it was not possible to fit the crystal structure to the EM envelope. Isn't this a concern?

The crystal structure of the MC1-Ypt7 construct only represents less than 30% of the complete MC1 complex. We tried to fit part of the crystal structure that corresponds to MC1 into the reconstruction. However, there were many different ways to fit the crystal structure into the reconstruction. At 17 Å and without a unique overall shape of the subcomplex, this is not surprising. We have included a 17 Å map of the crystallized MC1 subcomplex for comparison in the supplement to visualize this ambiguity (Supplementary Fig 4c).

There is no secondary validation of the EM map. Another low resolution method, SAXS for example, might be used to validate the EM envelope.

We respectfully disagree with the reviewer; electron microscopy is a self-contained method that does not need validation by another method. Especially not by SAXS, the resolution range of which is in the range of only 50 – 250 Å.

More importantly, there is no description in the manuscript of how the EM reconstruction benefits understanding of the biology of the complex. From my point of view, the manuscript has enough data without the EM, however if EM data is included in the manuscript, it needs to be better explained and more carefully integrated into the story.

The EM structure of MC1 clearly confirms, as suggested by gel filtration, that the complex represents a heterotetrameric assembly, thereby defining the principal architecture of the complex. It also represents the first overall structural description of MC1, and we believe the data should therefore remain part of the manuscript. We hope that after adding additional description and discussion of the EM data, their significance is better conveyed.

Some additional description and supplementary data are needed when MC1core is introduced. It would be beneficial to show the alignments from the different organisms as well as to illustrate the secondary structure elements. When core binding to Ypt7 is described there is no link to supplementary Figure 1d.

We have added sequence alignments of the crystallized parts of Mon1, Ccz1 and Ypt7 from *C. thermophilum* and *S. cerevisiae* and indicated the secondary structure elements as a Supplementary Figure 3. We also include the link to the gelfiltration profiles (now Supplementary Fig 1e).

The authors state that MtMC1full is a dimer based on the gel filtration profile, however the precise functional relevance isn't clear and this is not definitively established. Taking into account that MtMC1full has an elongated shape from the EM data, gel filtration cannot provide an accurate molecular weight of the complex. It would therefore be useful to add SEC-MALLS data including the molar mass distribution for all complexes used in the study if the authors have this.

We unfortunately do not have access to SEC-MALS at the University of Osnabrück or at the MPI in Dortmund. However, MC1 appears only slightly ellipsoid but mostly globular in the EM studies, and the MW determined based on the negative stain images corresponds to the apparent molecular weight determined on the calibrated size exclusion column. Also, the crystal structure shows that the MC1 core adopts a globular quaternary structure. Because of the consistency between SEC and structural data, we consider our molecular weight estimates highly reliable without additional experiments.

With regards to the crystallography described in the manuscript, both datasets are in the same space group and cell dimensions, in essence the same crystal form (including anisotropy) unless I have misunderstood something. However, the high-resolution data set has about 3000 fewer unique reflections compared to the low resolution (SAD peak) dataset and both data sets have 99% completeness. Is this correct?

The differences are due to the presence of pseudo-translational symmetry in the native, but not the derivative data set. Therefore, the high resolution dataset has less unique reflections.

Since the Rmeas values are relatively high, please add the CC1/2 values for both datasets.

CC1/2 values have been added to Table 1.

Reviewer #3 (Remarks to the Author):

The authors have determined the structure of the Ypt7/Rab7 guanine nucleotide exchange factor complex Mon1-Ccz1 (MC1 complex) at ~2.5Å resolution. The structure provides clues to features that allow discrimination of the Rab7-like Ypt7 over the Rab5-like Vps21 which are both located on the same membrane. The MC1 complex plays an essential role in a Rab cascade that defines the endolysosomal system and orchestrates the Rab switch between Rab5 and Rab7. The heterodimeric MC1 complex from *S. cerevisiae* has been successfully used in biochemical studies, but low expression levels, aggregation, and instability of the recombinant proteins have hampered structural studies. Authors have used Mon1 and Ccz1 from *Chaetomium thermophilum*, which show an analogous domain architecture compared to the *S. cerevisiae* MC1 complex. The full-length CtMC1 has been obtained and a variety of mutants produced to combine with the insight gained from the crystallographic structure of the catalytic CtMC1core in complex with Ypt7 to provide a significant advance in our understanding of the mechanism for this class of RabGEFs. The paper is recommended for publication. Some minor points need addressing:

1. Authors state "The globular shape of both the longin dimer and complete complex do not allow to fit the crystal structure of the core unambiguously into the EM density of full-length MC1" but offer no explanation,

We have included a 17 Å map of the crystallized MC1 subcomplex in comparison to the EM reconstruction of MC1 (Supplementary Fig 4c). Because of the overall globular shape and the relatively small portion (30%) that it represents of the full complex it could be fit at several positions. We can therefore not identify the position of the longin dimer in the full complex at this point.

2. Crystallographic data has somewhat limited completeness (86%), Wilson B is high etc. These should be commented upon (ideally a more complete data should be obtained). If not immediately possible then at least electron density for some regions of the structure should be shown.

Crystals of MC1coreYpt7 diffracted anisotropically, which prompted us to perform anisotropic correction (see methods). As a result of rescaling and truncation of the data, limited completeness is reported during refinement, but the original data collection had good completeness and redundancy (see "Data collection" statistics in table 1) and would not improve by collecting more data.

Anisotropy also resulted in high Wilson B factors for the original data, but was reduced after correction. We now also report this value. The quality of the maps obtained with this data was good, as can be seen in new Fig 3b.

REVIEWERS' COMMENTS:

Reviewer #1 (Remarks to the Author):

The authors have submitted the manuscript with minor revisions. Most of my comments have been dealt with to my full satisfaction. The manuscript must be accepted by all means.

As pointed out by reviewer #2 and myself, the EM-data are only a minor contribution to the structural and functional characterization of the complex structure. I therefore believe that these data are suited better for the supplement, as does reviewer #2.

It is advised to increase the font size of the atom labels in figure 7 since these are barely readable in their current form.

Reviewer #2 (Remarks to the Author):

Kuemmel and coworkers have provided a carefully revised manuscript that addresses all my comments on the initial submission. This is a very high quality study that will attract a considerable amount of interest, and I strongly support publication.

Reviewer #3 (Remarks to the Author):

Authors have addressed my reservations and have provided additional information. The manuscript can be accepted for publication as it provides a significant advance in our understanding of the mechanism for this class of RabGEFs

Response to reviewer #1 comments

Reviewer #1 (Remarks to the Author):

The authors have submitted the manuscript with minor revisions. Most of my comments have been dealt with to my full satisfaction. The manuscript must be accepted by all means.

As pointed out by reviewer #2 and myself, the EM-data are only a minor contribution to the structural and functional characterization of the complex structure. I therefore believe that these data are suited better for the supplement, as does reviewer #2.

We have moved all data on the EM analysis into the supplement as part of Supplementary Figure 1.

It is advised to increase the font size of the atom labels in figure 7 since these are barely readable in their current form.

We increased the font size of the atom labels in the schematic model (now Figure 6). Furthermore, we set the font-weight to bold to make the atom labels easily readable.

Reviewer #2 (Remarks to the Author):

Kuemmel and coworkers have provided a carefully revised manuscript that addresses all my comments on the initial submission. This is a very high quality study that will attract a considerable amount of interest, and I strongly support publication.

Reviewer #3 (Remarks to the Author):

Authors have addressed my reservations and have provided additional information. The manuscript can be accepted for publication as it provides a significant advance in our understanding of the mechanism for this class of RabGEFs